## Research Article

mental health; rural mental health; Maharashtra; mental health interventions; barriers in mental health care

**Corresponding author:**
Kaavya Iyer;
Email: kaavya@ananyabirlafoundation.org

# Mental health resources, barriers, and intervention needs among women in rural Maharashtra, India: A qualitative study

Kaavya Iyer[1] [ID], Pooja Gupta[2], Shivani Sapre[1], Tanvi Pawar[1], Pooja Gala[1], Hansika Kapoor[3], Rupa Kalahasthi[4], Arunima Ticku[1] [ID], Savita Kulkarni[5] and Poorvi Iyer[6]

[1]Ananya Birla Foundation, Mumbai, India; [2]National Institute of Mental Health and Neurosciences, Bangalore, India; [3]Monk Prayogshala, Mumbai, India; [4]Rochester Institute of Technology, Rochester, NY, USA; [5]Gokhale Institute of Politics and Economics, Pune, India and [6]London School of Economics and Political Science, London, UK

## Abstract

This research paper focuses on the mental health needs, the need for mental health interventions and barriers in mental healthcare of women living in rural Maharashtra, India. Using a mixed-methods approach, the study has collected data from a sample of women living in the rural areas of Maharashtra through in-depth interviews. The data collected has been analyzed to identify the barriers and obstacles in mental healthcare, how the existing community support serves as a resource as well as the desire for potential mental healthcare interventions among participants. The findings of the study are expected to contribute to the development of effective mental health interventions tailored to the specific needs of women living in the rural areas of Maharashtra. Overall, this research paper aims to improve the understanding of the mental health needs of women in rural Maharashtra and provide insights for policymakers and mental health practitioners to develop effective interventions to promote their mental well-being.

## Impact statement

This research study aims to understand current mental healthcare support and barriers experienced by women in rural India. By shedding light on this critical issue, this study aims to contribute to the development of targeted interventions that address the unique needs and challenges faced by these women. Additionally, on examining the existing support systems and barriers, the study identifies the gaps and challenges that prevent women from accessing adequate mental healthcare services. Ultimately, the findings of this research will contribute to bridging the gap in mental health care targeted toward women in rural India by ensuring equitable access to quality mental health support, overall improvement of mental health outcomes and the empowerment of women in rural communities.

## Introduction

According to the National Mental Health Survey (India State-Level Disease Burden Initiative Mental Disorders Collaborators, 2020), about 150 million Indians need mental health care and close to 10% suffer from common mental disorders such as depression, anxiety, emotional stress, suicide risk and substance use (Kallakuri et al., 2018). In 2014, India launched its first National Mental Health Policy and revised its Mental Healthcare Act in 2017. The objective of the act was to provide equitable, affordable and universal access to mental health care (Sagar et al., 2020).

Despite these provisions, there are notable gaps in the equitable access of mental health care across India. One such gap is seen predominantly among women, especially those belonging to rural communities. Distance, lack of infrastructure and lack of education and awareness have made it difficult to provide efficient, quality mental health services to women in remote villages in India.

A study by Gawai and Tendulkar (2019) suggested that married women in rural Maharashtra have little knowledge about mental health and are largely unaware of the mental health services that may be available to them. A community-based study by Srinivasan et al. (2020) has shown that about 15% of adult women in some rural villages of Southern India had experienced depressive symptoms and their risk factors were associated with discordant marital life such as living separately or being widowed and being less educated and that women who experienced domestic violence had significantly higher odds of reporting depressive symptoms.

In India, specifically in rural India, a number of barriers hinder the provision of mental health care and interventions. Gawai and Tendulkar (2019), in their study, highlighted that one of the key barriers for women seeking mental health services is stigma. They have identified stigma as a three-tier concept: lack of knowledge (ignorance and misinformation); negative attitudes (prejudice); and excluding or avoiding behaviors (discrimination). Self-stigma is the internalization of these negative attitudes. Their study emphasized that in rural India, strong negative beliefs and attitudes toward mental disorders underlie the need to improve mental health literacy among people. Therefore, the present study attempts to examine the mental healthcare services currently available, barriers that hinder seeking quality help and the desire and need for future interventions among women in rural Maharashtra.

## Objectives and background

According to Kallivayalil and Enara (2019), India has a population of 1.3 billion, and there is a significant lack of mental health personnel and infrastructure to cater to the needs of the people. They emphasized that India needs renewed strategies for rural community mental health care, to fill in this gap. A series of community- and clinic-based studies in India by Rao et al. (2011) have shown that women are twice as likely to be affected by depression as men and the potential factors that have contributed to this increased gender vulnerability are variables such as poverty, social class, marital and childbearing roles, lack of education and continued social oppression. They also found that women have been noted to not seek help because of stigma, poverty, paucity of awareness or access to care. Thus, their study found that psychological distress poses a major economic burden on poor women and adversely impacts their livelihoods.

With an increase in population, changing sociocultural values and lifestyles, frequent disruptions in income, crop failure, natural calamities (drought and flood), economic crises, unemployment and lack of social support, the number of people afflicted with mental health issues is expected to rise in rural areas (Kumar, 2011). Further, the statistics by the National Commission on Macroeconomics and Health (2010) showed that about 6.5% of the Indian population have some form of serious mental disorders, considering that almost 72.2% of the Indian population lives in rural areas, with only about 25% of the health infrastructure, medical personnel and other health resources, the gap between demand–supply of mental health care is highly disproportionate in rural areas (Kumar, 2011).

Sociological factors influence the subjective experience of people and therefore can impact mental health beliefs and, in turn, help-seeking behavior in rural areas. This is exacerbated by the lack of and decreased accessibility to healthcare facilities in rural areas (Chaturvedi, 2020). According to Shidhaye and Kermode (2013), the current treatment gap for mental illnesses is a result of inadequacy in the coverage of mental health services and underutilization of existing services by the community. They have stated that widespread stigma toward and discrimination against people with mental disorders is an important barrier against the utilization of existing services. This contributes toward a delay in seeking services and obstructs timely diagnosis and treatment for mental disorders, which further impedes recovery and rehabilitation. A study by Joag et al. (2020) showed that community-led interventions such as the Atmiyata project where one can use

volunteers from rural neighborhoods can prove to be a locally feasible and acceptable approach to reducing distress and symptoms of depression and anxiety in a low- and middle-income country context.

The establishment of primary health centers (PHCs) has, to some extent, aided in improving the affordability and accessibility of healthcare in rural India. However, these have largely been ineffective in addressing the needs of people suffering from or at risk of noncommunicable disorders including mental disorders (Kallakuri et al., 2018). Even if India is well placed with regard to trained personnel in general health services, mental health trained personnel are quite limited and based largely in urban areas (Khandelwal et al., 2004). Therefore, it will be interesting to observe and understand if organizations such as those aiding women through microfinance services (in the present study: Svatantra Microfin Pvt. Ltd. – A microfinance service designed to meet the needs of low-income individuals or small businesses who may not have access to traditional banking) act as a shared community resource for women, as it forms an integral part of their common lived experience.

Therefore, through this study, the aim is to (1) explore the gaps that exist between resources and prospective resources for the mental health of women in rural Maharashtra, (2) explore the role of existing community systems in the mental health of women in rural Maharashtra, and (3) to explore the plausible interventions to improve the mental health of women in rural Maharashtra.

Specifically,

*RQ1:* In rural Maharashtra, what gaps exist between available resources and prospective resources for the mental health of women?

*RQ2*: In rural Maharashtra, what is the role of existing community systems in the mental health aspect of women?

The present study aims to expand existing mental health literacy, which will further aid in curating or implementing pertinent mental health treatment or intervention plans.

Therefore,

RQ3: In rural Maharashtra, what are the plausible interventions to improve the mental health of women?

## Methodology

### Study design and participants

A qualitative cross-sectional study was conducted in Maharashtra, India, during the COVID-19 pandemic (October 2021–March 2022) using purposive sampling. The sample consisted of women entrepreneurs associated with Svatantra Microfin Pvt. Ltd., who had completed 2–3 loan cycles. The crux of the research was to focus on creating a sample of women from rural India. Hence, participants belonged to villages in the state of Maharashtra, India. To ensure variability, the sample was equally divided by age-group and caste status, and Svatantra branches with the highest number of clients were selected from each administrative division (Amaravati, Aurangabad, Nagpur, Nashik, Paschim Maharashtra and Konkan). Purposive sampling utilized caste and age as grouping categories to check for variability in understanding and manifestation of mental health conditions across age-groups as women of different ages may experience varying mental health concerns. Second, it has been noted that the caste system in India is one of the longest survival stratifications in the world (Thorat and Joshi, 2015). So, this would help the researchers understand if the social context of caste shapes contextual

**Table 1.** Caste- and age-wise distribution

| Age | General | OBC | Other | SC | Grand total |
|---|---|---|---|---|---|
| <35 years | 8 | 9 | 6 | 6 | 29 |
| 36–45 years | 4 | 6 | 5 | 5 | 20 |
| >45 years | 5 | 7 | 5 | 6 | 23 |
| Total | 17 | 22 | 16 | 17 | 72 |

understanding of phenomena related to mental health. As shown in Table 1, the final sample of 72 women ($M_{age}$ = 39.63 years) was selected from four caste groups (other backward class [OBC], general caste, scheduled caste [SC], and scheduled tribe [ST] [combined with other proportionately small castes]) (Vikaspedia, 2022) and three age-groups (<35 years, 35–45 years old, and >45 years), representing the most populous branches from all six administrative regions.

### Procedure

A standard conversational consent form was created. It entailed the following process: The interviewers introduced themselves and explained the purpose of the study. Consent was taken for participating in the study in audio and video, as well as audio recording of the call. The benefits and risks of participation were highlighted to each participant. Toward the end of the description, each participant gave an audio-recorded verbal consent.

For the qualitative exploratory study of mental health, a balanced sample of female clients was selected based on age, caste and region. Three interview modalities were used: in-person interviews, audio–video interviews at Svatantra branch offices and audio–video interviews conducted remotely. All interviews were audio recorded with consent. The interviews were done in the local language of the state of Maharashtra, that is, Marathi. The interviewers were early career professionals in the field of mental health with a Master in Clinical Psychology and native fluency in Marathi. The interview took around 30–40 min each.

Due to pandemic-related restrictions, the majority (56%) of interviews were conducted remotely, while 33% were done in person and 11% were conducted via audio–video at Svatantra branches. The audio was recorded across all modalities.

### Ethical considerations

Verbal informed consent was collected from the participants prior to data collection, and the participants were compensated for a day's wage (in kind through a mobile phone voucher worth INR 300; this was done at a later date). Those who had to travel to the Svatantra offices were compensated for their travel to-and-fro before they left the office. Approvals were obtained from the institutional review board at (MASKED FOR REVIEW) (Ref: #071–021).

### Reflexivity

Reflexivity is crucial when interviewing a mental health practitioner. As a team led by urban-dwelling women researchers, reflexivity in this context becomes crucial. Hence, the following aspects were taken into consideration.

Individuals in rural areas may struggle with mental health difficulties but lack the necessary words to describe their experiences. In contrast, urban mental health practitioners have access to resources and education that enable them to understand and articulate emotions and feelings.

Understanding instances of oppression and discrimination is another important consideration as urban mental health practitioners may have a better grasp of gender discrimination while rural women may accept gender discrimination as a way of life, lacking awareness of its impact or the need for change.

Income and financial independence also differ. In rural areas, women often lack financial knowledge and decision-making power, with husbands making all financial decisions. Urban women, on the other hand, tend to have a basic understanding of their family's finances.

Rural women's dependence on their husbands is evident in decision-making processes and even basic personal information being provided by husbands during interviews. In some cases, women are not allowed to have their own mobile phones. In urban settings, women enjoy greater decision-making autonomy and participate in family decisions.

Lastly, there is a stark difference in the treatment of genders in rural areas. Women occupy subordinate positions within the family, with major decisions made solely by husbands. Even if permitted to work, women are still responsible for family and household tasks. Widowhood can further exacerbate issues due to limited exposure to the outside world and lack of vital information.

Hence, understanding the context and unique challenges faced by rural women is crucial for mental health practitioners in order to provide appropriate support and intervention.

### Data analysis

Interview recordings were transcribed in English by a service provider and coded in Excel by two psychology master's degree holders who created the codebook. Responses were coded in Excel, with multiple codes given to responses covering various themes. Initial pilot coding of three interviews identified gaps and streamlined the coding process. Next, two pairs of fresh coders coded six interviews for consistency, with one coder replaced and trained in coding and the codebook. Inter-rater reliability was established through Kappa's coefficient, with discrepant responses discussed to arrive at a shared understanding.

### Coding procedure

A top-down coding approach was used to create a codebook, with researchers setting pertinent themes before beginning the analysis. The codebook had a global theme, divided into organizing themes with inclusion criteria and definitions. Organizing themes were a response to each corresponding research question tied to the global theme. In the process, inclusion criteria were reviewed and revised for unanimity among coders. Using a theory-driven approach, interview data classified within each organizing theme was coded using a bottom-up, inductive approach to build basic themes directly from the lived experiences of participants. The three global themes and their corresponding organizing themes are briefly described in Figure 1.

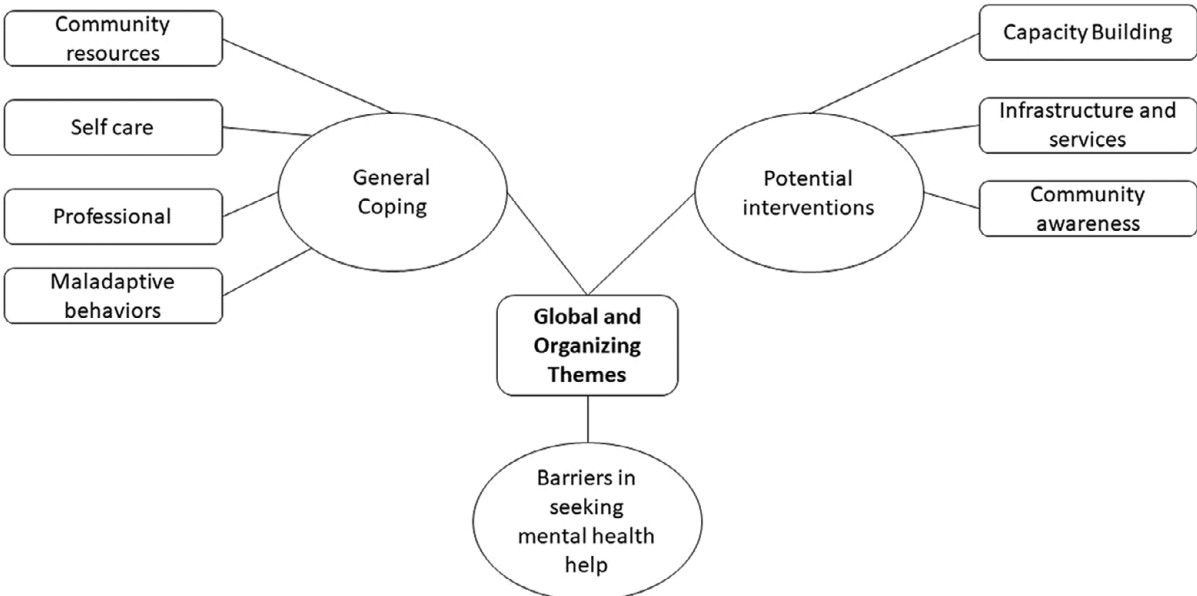

**Figure 1.** Global Themes and Corresponding Organizing Themes

## Results

### Demographic results

#### Personal interview

Most women were educated below the 10th grade (50%), followed by those educated till the 12th grade (16.6%), till the 10th grade (19.4%), graduation (4.16%) and post-graduation (2.7%). One woman each reported "other" and no schooling (1.38%). Thirty-eight women (52.77%) belonged to joint, and 34 (47.22%) to nuclear families, with an average of 5.5 people in the household. The women reported having an average of 1 child below 18 years and 1 child above 18 years.

Most women practiced Hinduism (82%), followed by Buddhism (15.2%), Islam (1.38%) and other (1.38%). Participants were recruited after matching their caste with the caste stratification in Maharashtra. Therefore, the sample comprised 27 (37.5%) women from OBC, 19 (26.38%) from SC, 10 (13.18%) from general, 7 (10%) each from ST and nomadic tribe and 2 (2.77%) from special backward class caste categories.

### Qualitative findings

#### Qualitative results

Below the major themes and their supporting quotations taken directly from the transcripts can be found. All the quotations were originally in Marathi; however, they have been suitably translated to English. Further, the only organizing theme with the greatest number of responses under each global theme has been described in this paper.

#### Global theme: General coping (RQ 1)

The global theme of general coping was created to understand what factors act as resources for mental health issues among women in rural Maharashtra and study the existing coping strategies these women use in order to cope with stressors.

*Community resources.* Utilizing existing community resources is an easy and accessible way of coping with daily stressors. It is also an informal way of seeking support as help is generally extended by friends, neighbors, "bachat gat" members and so forth. People who work for the community welfare are also considered to be authorities who can help deal with personal stressors. A common example of the same is Anganwadi/Asha workers.

We find that participants largely *rely on family members.* In a culture where mental health stigma is rampant, family members are often seen as safe and therapeutic sources of support. Coded responses show that most women resort to family members, especially husbands to deal with their stressors. One participant discusses.

> Extract 1: "My husband is very nice. If I need to express something, I express it before him."

Participants also discuss going to *authorities to discuss their issues.* After family, women in the villages go to Anganwadi/Asha workers to discuss their issues as these helpers are considered to be trained people by women.

> Extract 2: 'We regularly conduct these activities in Anganwadi and invite women over. Women tell us about their problems. We guide them about taking care of their newborns, their dietary needs, how to maintain personal hygiene, etc."

*Friends are also discussed as sources of support.* Participants stated that they talk to friends and women in the neighborhood to feel better.

> Extract 3: "They talk to their friends if they have any such problems."

#### Global theme: Potential interventions (RQ 3)

There is increasing recognition internationally of the need to build the capacity to strengthen mental health systems. This is a fundamental goal of the "emerging mental health systems in low- and middle-income countries" (Emerald) program, which is being implemented in six low- and middle-income countries (LMICs;

Ethiopia, India, Nepal, Nigeria, South Africa and Uganda) (Semrau et al., 2017).

The National Mental Health Program, which was launched in 1982 in India, was only partially able to fill in the treatment gap. Despite efforts to scale up the program from time to time, progress with the development of community-based mental health services and achievement of the desired outcomes in India has been slow. The main barriers include already overburdened PHCs, which face the following challenges: limited staff; multiple tasks; a high patient load; multiple, concurrent programs; lack of training, supervision and referral services; and nonavailability of psychotropic medications in the primary healthcare system. Thus, there is an urgent need to develop more community awareness, create relevant infrastructure and work toward capacity building to solve this problem (Hans and Sharan, 2021).

The aim of this global theme was to identify possible methods of interventions that can be employed to provide mental health aid to women in rural Maharashtra and was created to answer the research question – What kind of mental health interventions is/How can mental health interventions be catered to women in rural Maharashtra?

Capacity building.  Capacity building refers to strengthening skills and resources organizations and communities need to survive in a fast-changing world. This is an interesting finding as it could potentially imply that women in rural Maharashtra are cognizant of the lack of mental health resources and are open to receiving help. The rationale behind capacity building stems from the lack of resources, opportunities and an open environment where a need for better mental health care can be promoted and advocated. The global theme of potential interventions was created to answer the research question – What kind of mental health interventions is/How can mental health interventions be catered to women in rural Maharashtra?

A majority of participants cited a preference for group counseling. This indicates a greater comfort in a group setting that includes other women from the community, a consistent desire for women-led safe spaces and comfort and a willingness to share and seek mental health aid in group settings.

*Extract 4: Ma'am, it would be appropriate to talk to them in group. You would be able to explain to everyone. It is a big task to explain every woman personally."*

Women were enthusiastic about learned professionals coming to their towns and villages to talk about mental health and share knowledge and skills regarding psychological help and resources and, as such, indicated a desire for external help. Women in the communities want and would be receptive to mental health help provided by ABF (Ananya Birla Foundation) or other outside sources.

*Extract 5: "Yes, I mean if any outside woman comes, women may share their problems with her, they may hear her out. It would help them."*

In a direct and somewhat interesting contrast from the first basic theme, participants cited a desire for one-on-one, in-person counseling. This indicates desire for a private and safe space where there is no judgment, opinion or stigma from family or community member counseling.

*Extract 6: "Face to face to guidance or consultation. They will like it. It feels good if someone comes and talks about it one on one."*

## Global theme: Barriers in seeking mental health resources (RQ2)

Muhorakeye and Biracyaza (2021) stated that barriers to mental health interventions are more prominent in LMICs. These barriers to accessibility include stigmatization, financial strain, acceptability, poor awareness and sociocultural and religious influences. Hence, exploring the barriers toward seeking mental health services might contribute to mitigating them.

The global theme was created to answer the research question – What factors stop people from seeking mental-health-related help in rural Maharashtra?

Potential obstacles.  Potential obstacles refer to the existence of professional, financial and infrastructural barriers. A lack of such resources can be hindrances in seeking mental health help even if the community members are willing.

Participants stated that women in their communities would not be open to the idea of seeking help or learning more about potential resources that could aid in mental health help and psychoeducation. A lack of openness toward seeking mental health help and lack of mental health literacy in the community can lead to aversion toward seeking help.

*Extract 7: "Some women won't appreciate an outsider coming and won't let them talk".*

A lack of resources and a dearth of facilities or programs that provide mental health help present as a major barrier. There is a need and desire for mental health help within the community that is unfortunately met with a lack of opportunities or spaces to support them.

*Extract 8: "No, nobody like that comes to our village but there's a sarpanch in our village, and if any other women contest, then they can be approached and they make necessary arrangements but no one from outside comes here."*

*Lack of time acts as a barrier* due to household responsibilities and work commitments can serve as a deterrent in seeking help even if there is a desire to do so. Women in rural Maharashtra are burdened with taking care of large families and also (for the most part) earning a livelihood. These factors often act as a barrier due to the lack of time, energy and subsequently, motivation, in seeking mental health aid.

*Extract 9: "Usually no one has time. Most of us are always busy. They have a lot of work to do."*

## Discussion

The goal of this study is to better understand current mental health resources, barriers and needs among women in rural Maharashtra. The outcomes of this study have provided in-depth information that answer our research questions.

Results show that when it comes to coping with mental health concerns, women in rural Maharashtra use a number of existing resources such as community support, community helpers and workers and even family members as sources of support. This implies that some informal sources of social support act as pillars for mental health needs of women in rural communities. The study also focuses on two major facets of mental health help – barriers and desire for help. These two facets are interconnected and each influences the other. Barriers such as lack of time and financial resources were cited by most participants. This data can be taken into consideration when providing any potential interventions such as group counseling or individual counseling efforts.

The results show that the comprehensive and rich data collected has been instrumental in not only answering the research questions but also in providing potential avenues for future research. Additionally, in terms of infrastructure and services, participants have cited that they would like to have mental health services and mental health service centers built in the villages. Community awareness provided to the women in the villages by trained mental health professionals was also a preference cited by the participants. The study has achieved two broad objectives – understanding the current state of mental health needs, resources and concerns of women in rural Maharashtra and also understanding the desire for help as well as any precautionary measures that can be considered.

## Limitations

While an attempt was made to extensively cover varying facets of mental health concerns/needs among women in rural India, this study has a few limitations. Including participants from various rural regions in India would have provided more diverse and, hence, more generalizable data regarding the mental health needs of Indian women. Furthermore, by limiting the sample to only Svatantra beneficiaries, diversity within the sample population and the number of participants available were limited. Purposive sampling in this case can serve as a limitation. While an effort was made to include wide perspectives of those in the sample, the findings may not be generalizable to all women in rural India. Language served as a barrier in data collection, and hence, only Marathi-speaking participants from the selected administrative villages could participate. Finally, as data collection took place during the pandemic, a large number of participants were able to participate only via video calls or telephone calls as opposed to in-person interviews.

## Implications for future research

This study provides rich and valuable information that can help bolster not just future research but also mental health initiatives. Data from this study show that there is a desire among women from rural India to receive better mental health care; hence, future research could build on this information to determine what kind of care would be appropriate. Further, the study focuses on barriers and restrictions present in rural Indian communities that hinder mental health awareness and the provision of resources. This data can be used to anticipate barriers for initiatives that focus on mental health literacy and the provision of sustainable interventions.

## Conclusion

The goal of our study is to understand various aspects of mental health support available to women in rural Maharashtra. Our aim with this study is to investigate gaps that exist within current and prospective mental health resources for women in rural Maharashtra, explore the role of existing community support and explore the need for future mental health interventions.

With women from rural Maharashtra as our sample, we covered various research questions ranging from a need for better mental health support and existing mental health support to understanding barriers when seeking mental health help as well as solutions for possible interventions in the future.

The notion that mental health concerns exist among women from rural communities is well supported. Expanding on this literature, we conducted a qualitative study that focused on understanding the desire and need for mental health support among women in rural India. Results showed that among the participants, there is a desire to receive adequate mental health care. While literature regarding barriers to seeking mental health support and interventions exist, very few are tailored specifically toward women in rural India. As per the 2011 census, over 48% of the rural population in India comprises women. Results of our study showed that significant barriers, such as time constraints, stigma, daily responsibilities and lack of infrastructure, cause hindrances for women in rural communities from seeking mental health support.

The analysis opens further lines of inquiry. Our study shows that women in rural Maharashtra cited the desire to have easily available mental health resources. Conducting wide-scale tailored surveys could help delineate chief mental health concerns among women in rural Maharashtra and could facilitate the development of accessible and adaptable interventions. There is a desire within the community to end mental health stigma and gain access to mental health awareness and quality healthcare which sustains itself on wheels of cost effectiveness and easy accessibility and leads to better integration in the existing society structure. Interventions that could provide psychoeducation and are tailored to suit the needs of the community could prove to be not only helpful but also empowering to women in rural India.

This study can be used to open doors to future research that focuses on empowering, supporting and sustaining quality mental health initiatives and efforts for the betterment of women in rural areas.

**Open peer review.** To view the open peer review materials for this article, please visit http://doi.org/10.1017/gmh.2023.78.

**Data availability statement.** The data that support the findings of this study are available on request from the corresponding author. The data are not publicly available due to privacy and ethical considerations.

**Author contributions.** Conceptualization: K.I., T.P., P.Ga., H.K., R.K., A.T.; Data curation: K.I., P.Gu., S.S., P.Ga., A.T.; Formal analysis: K.I., P.Gu., S.S., T.P., P.Ga., P.I.; Investigation: K.I., P.Gu., S.S., P.Ga.; Methodology: K.I., P.Gu., S.S., P.Ga., H.K., R.K., A.T., S.K., P.I.; Project administration: K.I., P.Gu., S.S., P.Ga., H.K., R.K., A.T., S.K., P.I.; Resources: K.I., P.Gu., S.S., P.Ga., A.T.; Software: K.I., P.Gu., S.S., P.Ga., P.I.; Supervision: T.P., H.K., R.K., S.K., P.I.; Validation: S.K.; Visualization: K.I., T.P., P.Ga.; Writing – original draft: K.I., P.Gu., S.S., A.T.; Writing – review and editing: K.I., T.P., P.Ga., H.K., R.K., S.K., P.I.

**Financial support.** This research study was funded by the Ananya Birla Foundation.

**Competing interest.** The authors declare that there are no potential competing interests with respect to the research, authorship and/or publication of this article.

**Ethics approval.** This study was approved by the institutional review board at Monk Prayogshala (#084–022).

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
