## [Reviewer Report]

27 May, 2023

Gary Belkin,

Editor-in-chief

Cambridge Prisms: Global Mental Health

Dear Professor Belkin,

I am enclosing a submission to Cambridge Prisms: Global Mental Health, “Mental Health Resources, Barriers, and Intervention Needs among Women in Rural Maharashtra, India: A Qualitative Study”.

By using an underrepresented population consisting of women in rural Maharashtra, India, our paper has focused on understanding the current sources of mental health support available. Further, our paper has also studied barriers that these women face when seeking mental health support. Finally, our paper has studied the types and kinds of potential mental health interventions that the sample community would prefer in the future.

The notion that mental health concerns exist amongst women from rural communities is well supported, however, not much work has been done in digging deeper into the current state of support available and in understanding daily obstacles that a woman in rural India would face when seeking help.

Overall, this research paper aims at improving the understanding of the mental health needs and challenges of women from underrepresented populations. Most importantly, the study aims to serve as a foundation for future research and efforts in mental health interventions for minority populations. We believe that the results of this study will be of considerable interest to the readership of Cambridge Prisms: Global Mental Health.

I confirm that I do not have any conflicting interests that may be seen as influencing the research. I confirm that this manuscript includes original material that has not been published elsewhere and is not under consideration by another journal. I confirm that I have prepared the complete text suitable for anonymous review.

I confirm that all authors involved in this work have approved the submission of this manuscript. I will be serving as the corresponding author for this manuscript, on behalf of my co-authors, Tanvi Pawar, Shivani Sapre, Pooja Gala, Hansika Kapoor, Rupa kalahasthi, Pooja Gupta, Savita Kulkarni, Poorvi Iyer and Arunima Ticku. I look forward to hearing from you at your earliest convenience.

Sincerely,

Kaavya Iyer

Research Assistant, Project Blank slate,

Ananya Birla Foundation,

Level 27, Sunshine Tower,

Senapati Bapat Marg, Elphinstone Road, Mumbai-400013,

Maharashtra, India.

+919930218348 (voice)

kaavya@ananyabirlafoundation.org

---

## [Reviewer Report]

Review of the Article: Mental Health Resources, Barriers, and Intervention Needs among Women in Rural Maharashtra, India: A Qualitative Study

The reviewers acknowledge the effort of authors in assessing the Mental Health Resources, Barriers, and Intervention Needs among Women in Rural Maharashtra, the knowledge of which has implications in both clinical care service delivery and policy level. The reviewers would want to suggest a few critical points, which if, will make the article robust and comprehensive.

• Were all the research questions answered in this qualitative manuscript?

• It may be helpful if authors describe why the purposive sample was grouped according to caste and age? The justification would make more sense to the readers.

• Title: The title of the study conveys the theme on which the study is based on.

• As the study focuses on rural population, the same could be mentioned in the section on participants and design in methodology. Reviewers couldn’t find the mention of it in the section.

• The redundancy of description of Global theme under each sub section can be avoided in the section on Global Theme so that more can be conveyed in lesser words with due consideration to word limit.

• The authors could consider mentioning the organising and corresponding basic themes for Barriers in seeking mental health help in the Figure 1: List of global themes, organising themes and corresponding basic themes” which the reviewers felt is incomplete.

• Although the description of Global themes conveys the gist and summary, the reviewers felt that authors could describe the sub themes of- Infrastructure and services & Community Awareness in the description of results and discussion.

• The reviewers acknowledge the efforts by authors in studying the understanding of existing resources in the community. However, the reviewers would want to highlight existing programs such as Atmiyata Program which are running in the state of Maharashtra. Awareness about such currently existing resources and initiatives, if found in the results is worth mentioning. ( Joag K et al, 2020)1. This conveys the gap between the supply and demand which is important from both service delivery and policy perspective. If currently existing initiatives and their awareness and utilization were not part of the discussion, the authors could give it a mention in the section on limitations.

• Purposive sampling could also be mentioned as a limitation which maintains neutrality from scientific perspective.

• The reviewers feel that the statement on goal of the study in the conclusion can be same as the aims and objectives stated. It maintains coherency throughout the article.

Ref:

1. Joag, K., Shields-Zeeman, L., Kapadia-Kundu, N. et al. Feasibility and acceptability of a novel community-based mental health intervention delivered by community volunteers in Maharashtra, India: the Atmiyata programme. BMC Psychiatry 20, 48 (2020). https://doi.org/10.1186/s12888-020-2466-z

---

## [Reviewer Report]

Dear authors,

This is an interesting study and discusses an important angle of the needs specific to women living in rural areas of Maharashtra. However, the manuscript needs further details to make it more clear to the readers. Please find below the comments added section wise:

Introduction:

In the last paragraph, it is mentioned as ‘barriers that hinder seeking quality help’, how would you elaborate quality care? Especially considering rural settings, where there are already scare resources, it is important to know what the author means by this.

Objectives and background section

It will be good to elaborate a little about microfinance services, for larger audiences who might not be able to understand about this

Participants methodology- Not everyone will understand about the caste system in India. Please avoid abbreviations. e.g. OBC. Also please provide a reference to the constitution that describes the division of the caste system.

I think the consenting process needs to be detailed. I understand verbal consent was obtained, but was there a participant information sheet prepared? Was the purpose of the study indicated to the participants in detail? Were there any processes to record the verbal consent provided by the participant. What were the platforms used to consent since it was remotely done?

No details of the interviews given- how long each interview took, what language it was done? Who did the interview?

How was thematic saturation addressed?

Results- A table with the demographic details of the participants which include the number of interviews will be clear for the reader.

There is a mention about NMHP but reference not added?

In the obstacles section, it will be good to add any other quote that explains stigma in a better way. Currently it looks like an interpretation from the author

Would be good to add what kind of stigma exists, that might interfere with help seeking, are there any cultural or traditional things followed to deal with mental disorders which also might be considered while designing interventions.

Discussion- The discussion section needs to be improved.

Discuss how the findings of the current study corelate with any studies done in similar settings,

what did this study add to the previous literature already available on barriers. How can we develop interventions to get further information?

Since this study was done among women, have there been similar studies which focused on gender aggregated data and what have been the outcomes for the same?

---

## [Reviewer Report]

All the comments that have been made have been addressed. I recommend the current paper for publication.